# Usefulness of Hospital Admission Chest X-ray Score for Predicting Mortality and ICU Admission in COVID-19 Patients

**DOI:** 10.3390/jcm11123548

**Published:** 2022-06-20

**Authors:** Trieu-Nghi Hoang-Thi, Duc-Tuan Tran, Hai-Dang Tran, Manh-Cuong Tran, Tra-My Ton-Nu, Hong-Minh Trinh-Le, Hanh-Nhi Le-Huu, Nga-My Le-Thi, Cong-Trinh Tran, Nhat-Nam Le-Dong, Anh-Tuan Dinh-Xuan

**Affiliations:** 1Department of Diagnostic Imaging, Vinmec Healthcare System, Ho Chi Minh City 70000, Vietnam; v.tuantd5@vinmec.com (D.-T.T.); v.dangth@vinmec.com (H.-D.T.); v.cuongdm7@vinmec.com (M.-C.T.); v.mytnt@vinmec.com (T.-M.T.-N.); v.minhtlh1@vinmec.com (H.-M.T.-L.); v.nhilhh@vinmec.com (H.-N.L.-H.); v.myltn2@vinmec.com (N.-M.L.-T.); v.trinhtc@vinmec.com (C.-T.T.); 2Department of Respiratory Physiology, Cochin Hospital, AP-HP Centre, University of Paris, 75014 Paris, France; bacsinam81@gmail.com (N.-N.L.-D.); anh-tuan.dinh-xuan@aphp.fr (A.-T.D.-X.)

**Keywords:** chest X-ray, COVID-19, ICU admission, mortality risk

## Abstract

We aimed to investigate the performance of a chest X-ray (CXR) scoring scale of lung injury in prediction of death and ICU admission among patients with COVID-19 during the 2021 peak pandemic in HCM City, Vietnam. CXR and clinical data were collected from Vinmec Central Park-hospitalized patients from July to September 2021. Three radiologists independently assessed the day-one CXR score consisting of both severity and extent of lung lesions (maximum score = 24). Among 219 included patients, 28 died and 34 were admitted to the ICU. There was a high consensus for CXR scoring among radiologists (κ = 0.90; CI95%: 0.89–0.92). CXR score was the strongest predictor of mortality (tdAUC 0.85 CI95% 0.69–1) within the first 3 weeks after admission. A multivariate model confirmed a significant effect of an increased CXR score on mortality risk (HR = 1.33, CI95%: 1.10 to 1.62). At a threshold of 16 points, the CXR score allowed for predicting in-hospital mortality and ICU admission with good sensitivity (0.82 (CI95%: 0.78 to 0.87) and 0.86 (CI95%: 0.81 to 0.90)) and specificity (0.89 (CI95%: 0.88 to 0.90) and 0.87 (CI95%: 0.86 to 0.89)), respectively, and can be used to identify high-risk patients in needy countries such as Vietnam.

## 1. Introduction

The fourth wave of the coronavirus disease 2019 (COVID-19) epidemic in Ho Chi Minh City started with the first recorded case in a male patient admitted to Vinmec Central Park Hospital on 18 May 2021 [1]. According to the updated data, up to November 2021, this pandemic wave has caused the heaviest casualties in Vietnam with more than 440 thousand cases of COVID-19 infection in Ho Chi Minh City, and a mortality rate up to 3.85% [2]. Whilst infectious cases and mortality rates were very low in Vietnam during the first pandemic waves, in December 2021, the death rate from COVID-19 in Vietnam was 2%, equivalent to the worldwide 1.98% at that time [3,4].

Although chest X-rays (CXRs) have been used as a key diagnostic tool to document chest involvement in COVID-19 patients with dyspnea and/or hypoxia, chest radiographs alone are not recommended during the screening phase to diagnose COVID-19 pneumonia as findings on CXRs were deemed nonspecific with images that could overlap with those seen in non-COVID-19 infection [5]. Although they has a limited screening value in the general population in comparison to the reverse polymerase chain reaction (RT-PCR) [6,7,8,9,10,11,12], CXRs are often performed in patients hospitalized with COVID-19 due to their portability, wide availability, speed, and low cost. Furthermore, chest radiography is less associated with device contamination than computed tomography (CT), hence reducing exposure to healthcare workers. CXRs recorded on admission can therefore be used to assess the extent of lung disease prior to initiating any medical intervention that has the potential to establish the prognosis. Understanding the patient’s prognosis early at admission can in turn help guide patients and clinicians to the appropriate treatment. With a much higher frequency of use of chest radiographs than CT, radiologists can use radiographic findings to select patients who will likely benefit from intensive treatment, thus facilitating appropriate resource allocation right from the time of admission.

Thus, many developed predictive clinical models for COVID-19 outcomes in [13] often include age, comorbidities, and laboratory abnormalities and are rarely associated with CXR [14,15], though these imaging findings may provide prognostic information that cannot be obtained through other clinical and laboratory tests [16,17]. For the above reasons, the purpose of this study conducted at Vinmec Central Park Hospital was to evaluate the CXR scoring system’s ability, using chest radiographs performed at admission, to predict survival of COVID-19 patients.

## 2. Materials and Methods

We conducted a retrospective study on the records of all patients admitted to Vinmec Central from 26 July 2021 to 15 September 2021 with a positive RT-PCR test for SARS-CoV-2. This retrospective study was approved by our institutional review board, which waived the requirement for written informed patient consent.

Independent prognostic variables were collected, including epidemiological and clinical factors, such as age, sex, race, height, weight, date of admission, date of symptoms’ onset, current smoking status (defined as smoking within 1 year prior to admission), and chronic medical history, including: chronic obstructive pulmonary disease, diabetes mellitus, hypertension, coronary artery disease, cancer, pregnancy status, maximum temperature and maximum heart rate, maximum respiratory rate within 24 h of admission, capillary blood oxygen saturation (SpO_2_), patient’s oxygen requirements within 24 h of admission (ventilation, cannula, CPAP, HFNC, or intubation), treatment with Remdesivir and Molnupiravir, and vaccination status. To limit the confounding factors related to the treatment process, we only collected the first laboratory information within the first 3 days of admission, such as: blood count: neutrophiles, lymphocytes, platelets, hemoglobin, liver enzymes such as aspartate aminotransferase, and alanine aminotransferase.

Analysis of X-ray images within the first 24 h of admission: Digital X-ray images taken on the first day of admission were retrieved and anonymized as DICOM files, and images were interpreted using itk-snap software [18]. Three radiologists with varying years of experience (D.-T.T., T.-N.T.-M. and H.-D.T. with 6, 12, and >20 years), independently performed image analyses, and all were blinded to all clinical parameters except COVID-19 diagnosis in all patients. The CXR score [17] is assessed as follows: the two sides of the lung are divided into 4 regions based on two vertical lines between the middle and the horizontal line across the hilum on both sides. Extent score per area of lung parenchyma on chest X-ray: score 0–3 (0 = no lesion, 1 = less than one-third of area, 2 = one-third to two-thirds of the area, and 3 = more than two-thirds of the area). The severity score per area of lung parenchyma on chest X-ray (severity score) is assessed as follows: score 0–3 (0 = normal lung parenchyma, 1 = hyper-attenuated lesion does not obscure bronchial vascular structure, 2 = parenchymal hyper-attenuated lesions partially obscure bronchial vascular structures, and 3 = parenchymal hyper-attenuated lesions completely obscure bronchial vascular structures). The composite CXR score is the sum of severity and extent of 4 quadrants with equal weights. CXR score = ∑ extent score + ∑ severity score. Thus, no lung injury would be associated with 0 points, and the lesion covering the entire lung parenchyma would be associated with 24 points. Examples of AP radiographs and CXR reports of four patients with SARS-CoV-2 PCR-positive and different outcomes are shown in Appendix A.

The dependent variables include: The patient’s condition at the end of the hospitalization (survived, died); ICU status during hospitalization (no need, endotracheal intubation, dialysis, extracorporeal membrane oxygenation (ECMO)); classification of clinical severity during hospital stay according to Vinmec Healthcare System guidelines; and total number of days in hospital. Patients who had one of the following criteria were excluded from the cohort: those who were discharged within the first 24 H; those who had no chest X-ray data within the first 24 h; those who were intubated or died within the first 24 h; and those who were transferred to another hospital during treatment.

Statistical analysis: The 4-step statistical analysis process was conducted as follows: Step 1: evaluation of the agreement level among 3 different doctors regarding CXR score estimation by using Kendall’s coefficient of concordance (W) and corresponding significance test (Kendall and Gibbons, 1990); 1) survey on the degree of compatibility between CXR scores obtained from independent analysis results of 3 different doctors, based on Kendall’s W compatibility coefficient and chi-squared test according to Kendall and Gibbons (1990) [19]. This step is intended to check the authenticity and reliability of the CXR score data before they are used as the primary subject in further analyses. Step 2: exploratory data analysis: conventional descriptive statistics and data visualization were conducted to determine distribution characteristics of qualitative and quantitative parameters including CXR score among the subgroups of clinical outcomes (dead or survived, with or without ICU admission). Statistical inference on the difference in distribution of those parameters between groups was based on Mann–Whiney U test (for continuous data) and Pearson’s chi-squared test (for categorical data). Step 3: we performed an ROC analysis and a post hoc optimization process to determine the optimal cut-off points of the CXR score, allowing for prediction of death and ICU admission events. The effect of having high CXR scores on probabilities of occurrence of dead and ICU admission events was then estimated by non-parametric model Kaplan–Meier method [20] and log-rank test [19]. Step 4: time-to-event analysis was conducted to investigate the contribution of CXR score as a predictor of mortality and ICU admission. A time-dependent ROC AUC analysis was applied to evaluate the performance of the CXR score to make prediction of survival outcome at a given time point, in comparison with other routine clinical parameters. Next, we applied the “Select K-best” algorithm to determine the most efficient combination of 1 to 5 predictors which allow for accurate prediction of time to the occurrence of the event (death and ICU admission) through a Cox proportional hazard regression model, by optimizing the concordance index value of these models. For main statistical inference on the marginal effect of CXR score on the risk of death or ICU admission outcomes, we applied a Cox PH regression model [21] with CXR score and adjustment for patient’s age.

All statistical analysis was performed using Python programming language with the libraries pandas [22], scipy [23], lifelines [24] (survival analysis), matplotlib [25] (statistical graphics), and scikit-learn [26] (select K-best algorithm). Statistical inference was based on 95% confidence interval and null-hypothesis testing at significance threshold of *p* < 0.05.

## 3. Results

**Clinical characteristics of study population** 

Out of a total of 252 cases admitted to Vinmec Central Hospital from 27 July 2021 to 23 September 2021 with positive RT-PCR test results for SARS-CoV-2, 33 cases were excluded from the sample including 9 cases where no X-ray was taken within the first 24 h, 2 cases involved infants and pregnant women, 21 cases of incomplete medical records during the data collection period, and 1 case of ICU admission on the first day of hospitalization, so the final analysis was conducted on a dataset of 219 patients. Clinical characteristics of this cohort are presented in Table 1.

The mortality rate was estimated at 12.8% (28 patients died and 191 survived). About 15% of patients were admitted to the intensive care unit, and only 24% of them survived. The median in-hospital duration was 10 days, but it was significantly longer (19.5 days) for patients who died. Patients in the surviving group were significantly younger (median age of 52.7 vs. 69.15), but no difference was found for BMI. On admission, the vital signs, including breathing rate and pulse oxygen saturation, were significantly different between the two groups (*p* < 0.00001). A total of 58.9% of patients required supplemental oxygen, of which the majority of patients in the surviving group only breathed oxygen through a cannula 67/91, whereas the patients in the death group received high-flow oxygen (HFNC) or oxygen through a mask with a rebreather bag (26/28) within the first 24 h of admission.

Among the three most relevant comorbidities, including diabetes, hypertension, and cardiovascular disease, only diabetes showed a significant association with the death outcome in univariate testing. 

Comparative analysis of biomarkers obtained from blood tests on admission demonstrated that a higher quantity of neutrophil cells, a lower quantity of lymphocytes, and an increased value in CRP, ALAT, and ASAT were significantly associated with the occurrence of a fatal outcome.

**CXR score as a reliable measurement of lung lesions in COVID-19 patients** 

As described above, the CXR score for each patient was determined as the mean of estimations by three attending radiologists. Before analyzing CXR data, we needed to evaluate the inter-rater reliability of CXR score estimation by evaluating Kendall’s W coefficient of concordance. Comprehensive results of this evaluation are provided in Appendix A, and examples of clinical cases with X-ray images and corresponding CXR scores are presented in Appendix A data.

The levels of inter-rater agreement on assessing lung lesion extension were high for all four regions (from 0.772 to 0.794). Similarly, the agreement on severity was good for Q2 and Q4, and slightly lower but acceptable for Q1 and Q3. (W coefficient = 0.686 and 0.693, respectively). A very good agreement level was achieved for estimation of the total CXR score (W = 0.90, 95%CI: 0.886 to 0.919).

As shown in Figure 1 and Table 2, there was a clear contrast in distribution of CXR score values between two clinical outcome groups. The total CXR score and eight components were consistently and significantly higher for patients who died, in comparison with those of the surviving group. When stratifying by lung regions, lower parts show a higher extent and severity than upper regions. On a 0-to-3 scale, the left lower lung was the most severely damaged area, and the lower right lung was the area with the most widespread lesion in patients who died. The left upper lung was the part with the smallest extent in both groups of survivors and dead patients.

**CXR score as a predictor for the risk of mortality and ICU admission events** 

The results of ROC curve analysis in predicting death or ICU admission independently of the time factor (Appendix A) indicate that the CXR score could be a potential solution. The global predictive performance of the CXR score scale was excellent for both questions, with ROC-AUCs of 0.94 and 0.95, respectively.

At an optimal threshold of 16, the CXR score allowed prediction of death with a sensitivity of 0.91, a specificity of 0.86, and positive and negative predictive values of 0.87 and 0.90 (Appendix A). The same optimal cut-off could be applied for predicting ICU admission events, with very good performance (NPV of 0.87 and PPV of 0.88). A descriptive analysis using the Kaplan–Meier survival model and log-rank test (Appendix A) also confirmed that having a CXR score higher than 16 was significantly associated with a higher proportion of death and ICU admission events.

To evaluate the role of the CXR score alone or in combination with other potential clinical parameters in making predictions for time-dependent death and ICU admission events, we applied CoxPH regression analysis and the “selection K best predictors” algorithm. This process aimed to determine the most efficient combination sets of 1 to 5 predictors, allowing the corresponding CoxPH models to achieve optimal predictive performance (measured by concordance index).

For both prediction targets (death and ICU admission), CXR score was identified as the most important predictor, as the CXR variable participated in all best-performing models (which implies one, two, three, four, and five predictors). A significant effect of an increased CXR score value on the occurrence risk of death or ICU admission was found for all models (Appendix A).

Based on the results of this exploratory analysis, we developed a simple model with only two variables, CXR score and patient’s age, for statistical inference about the independent contribution of CXR score to the risk of death/an ICU admission event. The content of these two models is summarized in Appendix A. The results indicate a significant association between an increasing CXR score and higher risk of death (HR = 1.333, 95%CI: 1.097 to 1.621, *p* = 0.004) and ICU admission (HR = 1.641, 95%CI: 1.369 to 1.968, *p* < 0.0005). The marginal effect of CXR score levels ranging from 0 to 20 on the survival and ICU admission event function is visualized in Figure 2.

Furthermore, we compared the values of the time-dependent area under ROC curve (td-AUC) among 13 clinical parameters, including CXR scores, for predicting death at any time point within a follow-up period of 30 days. As shown in Appendix A and illustrated in Figure 3, the results indicate that CXR score is the strongest predictor within the first 10 days of admission. Its performance remained steadily good for a long period of 25 days (median = 0.853; 95%CI: 0.69 to 1.00). Other potential predictors include age (0.801, 0.637 to 0.932), AST (0.739, 95%CI: 0.638 to 0.932), and CRP (0.616, 95%CI: 0.406 to 1.00). After the 25th day of hospitalization, old age is the highest predictor of mortality.

## 4. Discussion

Study results for 219 patients hospitalized with COVID-19 at Vinmec Central Park hospital from July to September 2021 showed that the lung injury score for the chest X-ray could be a useful independent parameter with significant predictive power for mortality and hospitalization in the intensive care unit for patients with COVID-19. The high-risk CXR score, based on a cutoff of 16, provided 89% specificity and 90% sensitivity for predicting mortality, and nearly half of patients with high-risk scores died 30 days after hospitalization. With the above results, the CXR score can be applied to guide prognostic and early management decisions during the patient’s hospital stay.

We have developed clinical models to predict hospital mortality and ICU admissions. Many previous studies have shown worse outcomes in elderly patients with COVID-19 [5,14,15,27,28]. In our study, CXR score was the strongest predictor of mortality with a tdAUC of 0.85 (CI95% 0.69–1) during the first 15 days of hospitalization, and advanced age was the strongest predictor when the length of hospital stay increased over 3 weeks (tdAUC 0.78 95%CI 0.63–0.93) (Figure 3 and Appendix A). We also investigated the prognostic value of other paraclinical indices for the predictive model, including CRP, AST, and neutrophils, which had high mean predictive strength with a tdAUC of 0.59–0.72 (Figure 3 and Appendix A). Among the underlying diseases that influence prognosis, diabetes and obesity predicted mortality with a high tdAUC of 0.58–0.73 (Figure 3 and Appendix A). In the multivariable regression equation with a c-index up to 0.839, only the factors CXR score, liver enzymes, and AST had statistical significance (*p*: 0.002–0.04); background diabetes was not statistically significant at *p* #-0.107 (Appendix A, k = 3). When combined with the aging factor, the CXR score for the predictive model of mortality and critical course requiring ICU admission was statistically significant with a c-index of 0.825 and 0.925, respectively (Appendix A). The above results are in agreement with previous studies on the predictability of radiographic lesions in COVID-19 pneumonia, although the previous authors did not incorporate laboratory indices in their model [27,29]. Previous studies that used CXR to predict outcomes in COVID-19 patients did not include laboratory indices [14,27] or were limited by a small sample size [30]. Some studies on the role of biological indicators recorded leukopenia and severity of lung injury on X-rays to determine their relation to the need for supplemental oxygen [30]; elevated and lymphocytes neutrophil ratio are associated with an increased risk of critical illness [10,15], and LDH is associated with disease severity [15] or the need for oxygen support [30].

Accurate and easy-to-apply scoring criteria are essential for a clinically useful prognostic scale. Various radiographic methods of severity have been developed to quantify the severity of lung disease in tuberculosis [31], severe acute respiratory infection [32], respiratory distress syndrome, and respiratory acute distress syndrome [33], and COVID-19 [10,27,34]. In our study, the lower lung area showed greater severity and spread than the upper lung area. This lower lobe predominance of COVID-19 has been described previously [11,12,14,27]. The scoring criteria used were suggested by Reeves et al. [17] and are similar to the Brixia score, although the Brixia score divides the lungs into six regions rather than four [34]. In addition, the Brixia score does not refer to the severity of COVID-19 pneumonia, while our composite CXR criteria consider both the extent and severity of the pneumonia lesion on imaging. We found a higher consensus among physicians for the CXR score (κ = 0.90; 95%CI, 0.89–0.92) than previously recorded results for the Brixia score (κ = 0.82; 95%CI, 0.79–0.86) [34]. Our CXR severity score indicates good reliability for radiologists with 6, 12, and over 20 years of experience, which was not reported in studies using Brixia points. Taylor et al. [32] observed high confidence among radiologists, clinical general practitioners, residents, nurses, and medical students for a simple classification system of only six severity levels for acute respiratory distress syndrome. Therefore, it is necessary to conduct extensive research on the assessment of CXR severity in different subjects from many different disciplines and training levels to ensure the widespread availability of the scale.

Due to the high mortality rates observed in patients with COVID-19 pneumonia, it is important to identify patients at risk. Toussie et al. ([27], p. 19) and Hui et al. ([30], p. 19) observed that CXR score severity was an independent predictor of intubation, although it was not associated with in-hospital mortality. However, in the study by Hui et al., the sample size was too small (*n* = 19) to predict mortality, and the study by Toussie et al. excluded patients over 50 years of age who belong to the highest-risk subgroup [27,30]. Cozzi et al. [14] observed that CXR severity was associated with an increased risk of ICU admission, but they did not evaluate patient mortality outcomes. Borghesi et al. [29] observed a relationship between CXR severity and hospital admission mortality using an 18-point severity scale, which is 6 points lower than the 24-point range used in our study. These authors reported a mortality comparable to our results for each 1-unit increase in CXR severity on multivariable regression (OR, 1.33; 95%CI, 1.20–1.47). In our study, each point increase in the CXR score increased the risk of death over time, HR, 1.33 times (95%CI, 1.10–1.62) (Appendix A). In the study by Reeves et al., with the same CXR score, each 1-unit increase in the composite CXR score was associated with a 17% increase in mortality (OR, 1.17; 95%CI, 1.1–1.24; *p* < 0.001). The difference in likelihood is due to differences in the status of patients admitted to different facilities. In the US, during the period of March and April, 2020 at the peak of the epidemic, only very serious patients were hospitalized; in contrast, at our institution, in August and September 2021, serious patients were mainly picked up from field treatment centers. The rest were patients who had come to the hospital on their own with varying degrees of severity. 

Liang et al. [15] observed that any CXR abnormality was associated with an increased incidence of critical illness (OR, 3.39; 95%CI, 2.14–5.38), and this criterion was an independent predictor in their multivariate model. However, their binary scoring methods limit the differential risk assessment of patients with different degrees of disease severity [15,30]. In our study, the CXR severity score on admission was an independent predictor of mortality and critical progression requiring ICU admission. The results provide further insight into the role of CXR in predicting clinically meaningful outcomes, including mortality and progression in COVID-19 patients.

Many previous studies have explored the severity classification of COVID-19 pneumonia using computed tomography (CT) [35,36,37] However, unlike chest radiographs on admission, CT is not routinely used. Studies using CT for reclassification included a disproportionate number of patients who were imaged to assess for clinical worsening or to exclude other reasons for hypoxemia, such as pulmonary embolism. Therefore, CT is not as easily applicable to the prognosis of COVID-19 at the time of admission as chest radiographs, which are widely available, inexpensive, and, in the case of portable chest radiographs, easier to decontaminate [38,39,40]. Lung ultrasonography has also been documented as a diagnostic modality for COVID-19 pneumonia, with at least one study showing that lung ultrasonography is more specific and sensitive than chest radiographs [41]. However, ultrasonography is highly operator-, and technician-dependent, and evaluation of the entire lung is difficult due to the limited propagation of ultrasound waves through the air tissues. We therefore believe that chest radiographs are the ideal imaging modality for determining the severity of COVID-19 pneumonia and provide useful prognostic information for clinicians’ requirements.

Despite the positive results provided by our study, it also has limitations. First, our results were obtained from a single center in Ho Chi Minh City with relatively adequate facilities that may not be representative of patients from other regions. Second, patients were retrospectively identified based on known COVID-19 infection. Our chest X-ray severity score is not designed to differentiate between COVID-19 pneumonia and other lung diseases. Therefore, the results of our scoring system should only be applied to confirmed cases of COVID-19 by RT-PCR. However, CXR severity scores on admission can be obtained pending RT-PCR results because of the quick and easy nature of this imaging facility. Third, our CXR severity scoring system was used only for anterior or posterior radiological evaluation. The findings on the lateral radiograph may provide additional information that can be used for more precise classification. We did not evaluate lateral radiographs because most patients in our hospital system with confirmed or presumptive COVID-19 received anteroposterior radiographs only. Fourth, previous studies have shown that CXR severity peaks 10–12 days after symptom onset [11]; we found that the value of X-ray over time was highest at about 5–10 days after admission (tdAUC chart), then gradually decreased but still remained high compared to other tests. At the same time, in patients with a long hospital stay, age proved to be a strong prognostic factor, so we built a predictive model based on two factors: CXR score and age. Instead of building a model based on the disease stage, we designed a model based on the time of admission to ensure uniformity and informational utility for clinicians. Finally, the model did not include patients with comorbidities other than the underlying medical condition and did not compare other clinical risk scores such as the 4C score [42] due to the limitation of retrospective study design with limited available data. Future research directions include carrying out a prospective design and expanding the study population to be able to verify the validity of the established model.

## 5. Conclusions

In this study, we investigated a simple scoring system for grading COVID-19 severity that could be used by radiologists with varying years of experience working in hospitals where HRCT is not readily available. Chest radiograph severity score on admission was a significant independent predictor of mortality and hospital ICU admission. The CXR score had good sensitivity and specificity and can be used to identify high-risk patients. We combined additional CXR scores, epidemiological factors, underlying disease, and blood tests to build multivariable regression equations predicting hospital outcomes such as mortality and admission to ICU. The results can be applied to guide risk assessment and clinical decision making at the time of initial contact with a patient with COVID-19.

## Figures and Tables

**Figure 1 jcm-11-03548-f001:**
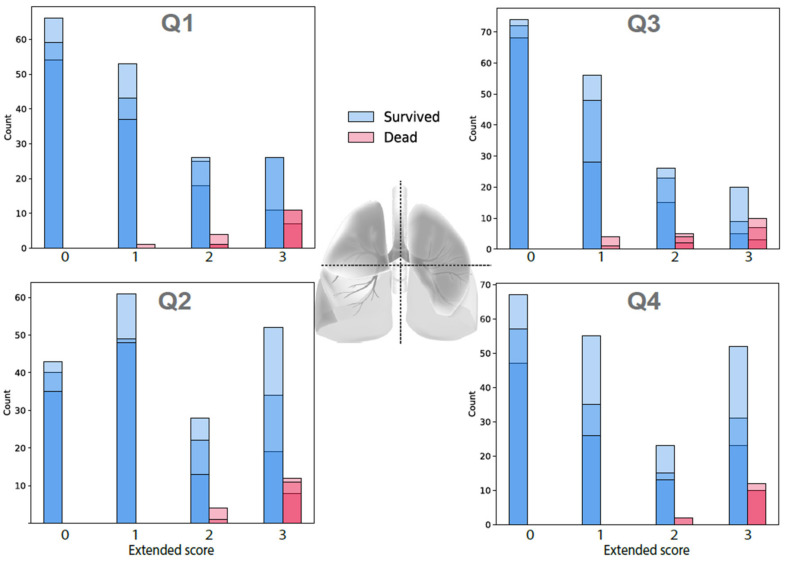
Distribution of CXR score values in surviving and dead groups. The blue and red color different was marked by the scoring of each one among 3 radiologist participant.

**Figure 2 jcm-11-03548-f002:**
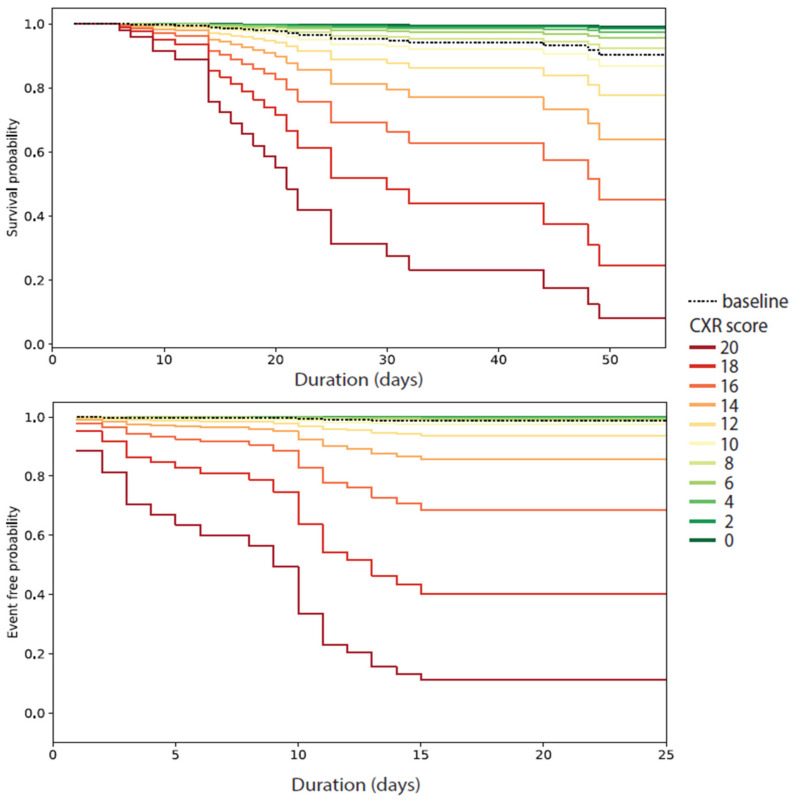
The effect of CXR score ranging from 0 to 20 on the survival and ICU admission event function. Note that only in-hospital mortality was taken into account.

**Figure 3 jcm-11-03548-f003:**
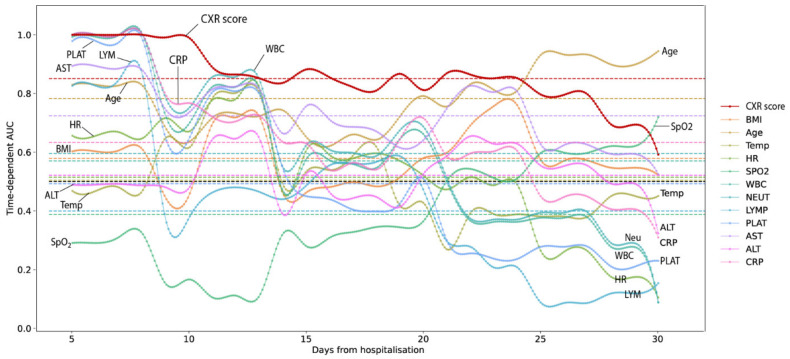
Time-dependent AUC for predicting death within 30 days among 13 clinical parameters, including CXR scores.

**Table 1 jcm-11-03548-t001:** Clinical characteristics of study population.

Variable	Pooled Data	Survival Outcome Subgroups		
		Survived	Died	*p* Value *
Number of patients	219	191	28	
Age, year	57.77 (18.11–81.29)	52.72 (17.07–78.73)	69.15 (52.77–83.86)	<0.00001
Male gender: freq (%)	111/219 (50.68)	97/191 (50.75)	14/28 (50.00)	1.0
Body mass index, kg/m^2^	23.44 (17.86–30.04)	23.31 (17.8–29.78)	24.09 (19.77–29.52)	0.04
Admission measures				
- Temperature, °C	37.0 (36.5–38.5)	37.0 (36.5–38.5)	37.00 (36.4–38.6)	0.41
- Respiratory rate, breaths/min	20.0 (18–31.75)	20.0 (18–29)	26.00 (20–41.4)	<0.00001
- Heart rate, beats/min	88.0 (68.95–115)	88.0 (68.55–113.9)	90.5 (71.35–120)	0.149485
- Pulse oxygen saturation (SpO_2_) %	96.0 (83.4–98.0)	96.0 (86.4–98)	89.0 (70–97)	<0.00001
Oxygen support within first 24 h				
Room air	90 (41.10%)	90	0	<0.0001
Canula	69 (31.51%)	67	2	
HFNC	27 (12.33%)	18	9	
Mask	23 (10.05%)	14	9	
NIV	6 (2.7%)	1	5	
Mechanical ventilation/Tracheal intubation	4 (1.8%)	1	3	
Blood test and other biomarkers				
White blood cell count, 10^9^/L	6.4 (3.71–14.09)	6.10 (1.9–10.63)	9.10 (3.31–21.17)	<0.01
Neutrophil count, 10^9^/L	4.6 (1.9–12.8)	4.30 (1.9–10.66)	8.50 (3.31–21.17)	<0.0001
Lymphocyte count, 10^9^/L	1.0 (0.3–2.4)	1.10 (0.325–2.5)	0.60 (0.2–1.57)	<0.0001
Platelet count, 10^9^/L	204 (115–401)	202.0 (115.3–383.7)	205.0 (96–431.5)	0.486075
Hemoglobin, g/dL	13.20 (10.2–15.9)	13.25 (10.13–15.96)	13.10 (11.13–15.62)	0.296964
Aspartate Aminotransferase, U/L	39.0 (17.55–137.35)	34.55 (16.96–118.98)	83.70 (36.73–213.88)	<0.00001
Alanine Aminotransferase, U/L	37.45 (12.37–132.03)	33.20 (11.54–121.38)	67.20 (26.87–212.29)	<0.0001
C reactive protein, mg/L	23.52 (0.93– 161.75)	18.40 (0.84–150.62)	86.71 (26.81–169.26)	<0.0001
Comorbidities				
Diabetes	79/219 (36.07)	66/191	13/28	<0.001
Hypertension	75/219	60/191	15/28	0.09
Cardiovascular disease	15/219	10/191	5/28	0.04
Antiviral treatment				
	154/219	129/191	25/28	0.042
Remdecivir	152	127	25	
Molnupiravir	2	2	0	
COVID–19 vaccination				
Dose 1	107/219	98/191	9/28	0.163838
Dose 2	6/219	6/191	0/28	0.518534
Severity degree on admission				
1	90 (41.10%)	90	0	<0.0001
2	61 (27.85%)	57	4	
3	65 (29,68%)	43	22	
4	3 (1.37%)	1	2	
Most severe degree during hospitalization				
1	68 (31.05%)	68	0	<0.0001
2	63 (28.77%)	63	0	
3	53 (24.20%)	53	1	
4	34 (15.53%)	7	27	
Intensive care unit admission,	34 (15.28%)	8/191	26/28	<0.00001
- Mechanical ventilation	29	7	22	<0.00001
- CRRT	16	1	15	<0.0001
- ECMO	3	0	3	0.0002
Length of symptom onset, day	16.0 (9–51.6)	15 (9–35.5)	27.5 (12.7–56)	<0.00001
Length of stay, day	10.0 (4–35.9)	10.0 (4–28)	19.5 (7.7–46.95)	<0.00001

Note: quantitative data are described as medians (5th–95th percentiles); categorical data are summarized as frequencies (%) for total number of patients with available data for each subgroup: ECMO = extracorporeal membrane oxygenation; CRRT = continuous renal replacement therapy; HFNC = high-flow nasal cannula; NIV = noninvasive ventilation. * *p* value: for quantitative variables, Mann–Whitney U test was used to test the null hypothesis that the 2 subgroups have a uniform distribution; for qualitative/discrete variables, the chi-squared test was used to test the null hypothesis of the independence between the two groupings.

**Table 2 jcm-11-03548-t002:** Chest radiograph scores on the first 24 h from admission among surviving and dead patients.

	Pooled Data	Survival Outcome		
		Survival	Died	*p* Value *
Size	219	191	28	
CXR score	9.33 (12.33)	7.33 (11.33)	19.67 (2.33)	<0.0001
Extent Q1	3.0 (6.0)	3.0 (5.0)	9.0 (1.5)	<0.0001
Extent Q2	4.0 (6.0)	3.0 (6.0)	9.0 (1.0)	<0.0001
Extent Q3	2.0 (5.0)	2.0 (4.0)	7.0 (7.0)	<0.0001
Extent Q4	4.0 (7.0)	3.0 (6.0)	9.0 (1.0)	<0.0001
Severity Q1	3.0 (4.0)	3.0 (3.0)	6.0 (1.0)	<0.0001
Severity Q2	4.0 (4.0)	3.0 (4.0)	6.0 (0.0)	<0.0001
Severity Q3	2.0 (3.5)	2.0 (4.0)	6.0 (2.0)	<0.0001
Severity Q4	3.50 (5.0)	3.0 (4.0)	6.0 (1.0)	<0.0001

* Note: data are described as medians (interquartile range); * *p* values are based on Mann–Whitney U test.

## Data Availability

Link to publicly archived datasets analyzed: https://docs.google.com/spreadsheets/d/1kWnufPuE-tjOOvgl8k1imECUNI0GviLA/edit?usp=sharing&ouid=103358838265956753801&rtpof=true&sd=true, accessed on 20 April 2022.

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
