# Peer review of "Usefulness of Hospital Admission Chest X-ray Score for Predicting Mortality and ICU Admission in COVID-19 Patients"

_jcm, 2022, doi:10.3390/jcm11123548_

Round 1

Reviewer 1 Report

The present manuscript embodying usefulness of chest X-ray score for the evaluation of severity in COVID-19 patients is a statistical model. While RT-PCR is the globally accepted COVID-screening method, CXR is largely underrepresented due to its non-specific nature and is not a primary diagnostic option. One detects presence of pathogen and the other detects pulmonary severity. The perspective is, if combined together, we can not only provide better treatment using predictive mode but, most importantly, can save lives. The present work is a significant step towards this perspective. The cohort is well inclusive of many variables such as SpO2, blood count, COVID-related symptoms, liver functions etc. These possibly make this work a useful guide to medical providers. 

The usefulness of hospital admission chest X-ray score for predicting mortality in COVID-patients is adequately reported and well supported by statistical analysis. Hopefully, this will help the medical community. 

Author Response

We thank you for your comments and the time devoted to read and assess our manuscript.

Reviewer 2 Report

In this interesting article, a chest x ray scoring system for COVID-19 severity is evaluated. It is shown to be a useful predictor for death/ICU admission. The manuscript is overall clearly written and of interest for the scientific community. I do not have major concerns. however, I have minor comments that will improve the overall quality of the work.

  • language:
    • improvement of English/spelling errors before publication is needed. e.g. page 3, line 108, page 5 line 168-171: this sentence is difficult to understand.
    • rather than "disease onset", I would refer to "symptom onset" - Table 1, second last line, figure S1 legend
  • methodological concerns:
    • how was the survival up to 70 days calculated? was there a follow up visit? if yes, please specify in methods. in this case, it seems surprising to me that the institutional review board waved an informed consent. if not, were only patients still in hospital included in the survival analysis? in this case, please clarify in text/ figure titel (figure S2/Figure 2) that in-hospital mortality is taken into account
    • the number of "death" events for Cox regression models is 28. this seems low for testing up to 5 independent variables. can you justify this?
    • p. 3 l. 134: "in comparison with other routine laboratory parameters". as far as I see, age and diabetes are also included as variables - "laboratory parameters" should be replaced by "clinical parameters".
  • results:
    • clinical characteristics: no information on vaccination status is given. this should be added even if the cohort only includes unvaccinated individuals/no information is available as it may tremendously change outcome.
    • most relevant comorbidities: the authors state that only diabetes, but not other comorbidities showed an association with death. please add "in univariate testing" as no multivariate testing has been performed
  • Tables and figures:
    • Figure 3: even on a big screen, it is very difficult to distinguish colours of parameters other than CXR_score, e.g. to distinguish BMI from temperature. this should be improved.
    • Figure S2: why does the specificity curve (in blue) only start at CXR score 10?
  • discussion/conclusions: In the abstract, the authors claim that the CXR score presented here is suitable for low-income countries like Vietnam. this is a valuable approach. however, for a clinician it would be interesting to see if CXR scoring at admission performs better than simple and widely used risk scores at admission such as 4C score that is also appropriate for similar settings. introduction and discussion section should be updated with relevant clinical risk scores , e.g. https://doi.org/10.1136/bmj.m3339), https://doi.org/10.1038/s41598-021-97332-1. as far as I see, all variables for the 4C score (maybe with exception of GCS) should be available in this cohort. it would be interesting to see a comparison between this risk score and the models proposed here.

Author Response

We thank you for your comments and the time devoted to read and assess our manuscript. Please find below our point-by-point response to the comments.

-------------------------------------------------------------------------------

In this interesting article, a chest x ray scoring system for COVID-19 severity is evaluated. It is shown to be a useful predictor for death/ICU admission. The manuscript is overall clearly written and of interest for the scientific community. I do not have major concerns. however, I have minor comments that will improve the overall quality of the work.

  • language:
    • improvement of English/spelling errors before publication is needed. e.g. page 3, line 108, page 5 line 168-171: this sentence is difficult to understand. -> corrected
    • rather than "disease onset", I would refer to "symptom onset" - Table 1, second last line, figure S1 legend  -> corrected
  • methodological concerns:

    • how was the survival up to 70 days calculated? was there a follow up visit? if yes, please specify in methods. in this case, it seems surprising to me that the institutional review board waved an informed consent. if not, were only patients still in hospital included in the survival analysis? in this case, please clarify in text/ figure titel (figure S2/Figure 2) that in-hospital mortality is taken into account

à We thank you for your suggestion that we need a clearer description of the survival analysis design, showing your careful analysis of the data and how they have been generated.

In our study, the time-to-event analysis is established as follow: death and ICU admission are two target events; the observation period is limited by the time of hospital admission and occurrence of events or censoring. Patients are monitored as long as they remain at the hospital, thus the survival time could be longer than 30 days for some cases. Data collection is retrospective, and we did not perform posterior follow–up visit. There were 3 possible causes of follow-up disruption: patient was transferred, voluntarily discharged, or recovery, these outcomes are considered as “censoring”. Only in-hospital mortality was taken into account.

    • Cox regression models is 28. this seems low for testing up to 5 independent variables. can you justify this?

à We thank the reviewer for this important remark. We agree that the number of events per variable (EPV) may have an impact on the accuracy of statistical inferences on regression coefficients, confidence interval and statistical power. However, as showed in the figure below, the impact is complex and depends on the nature of each covariate (continuous or binary, prevalence and effect size).

Figure: Patterns of relationship between statistical power and number of events per variable in a proportional hazard survival regression model, based on simulated data.

Therefore, it would be hard to justify whether 28 events would be appropriate or not for a Cox-PH model that included 4 or 5 covariates. In practice, we usually apply a simple rule that EPV = 10 is sufficient to ensure accurate estimation (if all covariates are continuous or binary with moderate prevalence).

In our study, the main statistical inference was based on a Cox-PH model with only 2 covariates (Age and CXR score, both are continuous, with a strong effect of CXR score). As the models with 4 and 5 covariates were simply provided as supplemental results, we decided to remove those results in the new version. Thus, experiment of finding the best variable combination will be limited at k=3.

    • p. 3 l. 134: "in comparison with other routine laboratory parameters". as far as I see, age and diabetes are also included as variables - "laboratory parameters" should be replaced by "clinical parameters".à. -> corrected
  • results:
    • clinical characteristics: no information on vaccination status is given. this should be added even if the cohort only includes unvaccinated individuals/no information is available as it may tremendously change outcome. 

à We have added the vaccine information to the new version, but the type of vaccine and length from vaccination to symptom onset was not available

 In July 2021, the majority of people in Saigon only had one first dose of vaccine covid

Vaccine

Dose 1

107/229

98/201

9/28

0.163838

Dose 2

6/229

6/201

0/28

0.518534

. This information has been included in the new version of analysis plan.

    • most relevant comorbidities: the authors state that only diabetes, but not other comorbidities showed an association with death. please add "in univariate testing" as no multivariate testing has been performed à corrected
  • Tables and figures:

Figure 3: even on a big screen, it is very difficult to distinguish colours of parameters other than CXR_score, e.g. to distinguish BMI from temperature. this should be improved.

We agree on this point. We tried to improve the readability of Figure 3 by adding labels at two bounds of the curve for each parameter.

    • Figure S2: why does the specificity curve (in blue) only start at CXR score 10? 
    • à this was due to the identical values of specificity (blue color) and Youden’J index (purprle color) for CXR values within the interval of [0 – 12] (absolute sensitivity was reached within this range).

  • discussion/conclusions: In the abstract, the authors claim that the CXR score presented here is suitable for low-income countries like Vietnam. this is a valuable approach. however, for a clinician it would be interesting to see if CXR scoring at admission performs better than simple and widely used risk scores at admission such as 4C score that is also appropriate for similar settings. introduction and discussion section should be updated with relevant clinical risk scores , e.g. https://doi.org/10.1136/bmj.m3339), https://doi.org/10.1038/s41598-021-97332-1. as far as I see, all variables for the 4C score (maybe with exception of GCS) should be available in this cohort. it would be interesting to see a comparison between this risk score and the models proposed here.

We thank the reviewer for these suggestions. Unfortunately, some components of the 4C score were not available in the present cohort (such as Glasgow coma scale and urea level).[1] We will consider this as a limitation of our study and refer the alternative tools as the reviewer mentioned in our discussion.

[1]          S. R. Knight et al., “Risk stratification of patients admitted to hospital with covid-19 using the ISARIC WHO Clinical Characterisation Protocol: development and validation of the 4C Mortality Score,” BMJ, vol. 370, p. m3339, Sep. 2020, doi: 10.1136/bmj.m3339.

Reviewer 3 Report

1. Objectives and need of the study can both refined 

Material and methods 

2. Why didn't the authors think of stratification using RT-PCR scores 

3. A lot of data has been conveyed which look superfluous , Kindly rearrange with paragraphs and continuity with each other 

4. Same applies to statistical analysis 

5. Do you think a flowchart can be added so that lot of information can be transferred there and reduce tables.

6. Few information about parameters of mechanical ventilation is missed like PEEP, mode ? used, FiO2 maintained during the process , patients who used MV for long are more prone for severe scores ?

7. Discussion 

Reduce the amount of words in Discussion Please discuss findings only 

Divide into paras so that they are easy to read 

Limitation to study ??

Were the results compared To MRI ,CT scores as it remains the Gold standard to check the severity of lung 

Author Response

We thank you for your comments and the time devoted to read and assess our manuscript. Please find below our point-by-point response to the comments.

-------------------------------------------------------------------------------

Reviewer 3:

  1. Objectives and need of the study can both refined 

à These two sections have been rewritten

Material and methods 

  1. Why didn't the authors think of stratification using RT-PCR scores 

à Even though association between RT-PCR and chest CT scores has been observed in some recent studies, we did not consider including RT-PCR score in our data analysis, since we cannot ensure a standardized protocol in which both chest X-ray (CXR) and RT-PCR sampling should be performed at the same time for all patients. In some patients, the RT-PCR test has been conducted by the community healthcare service several days before hospital admission. Furthermore a RT-PCR positive result would only imply infection by SARS-CoV-2 which does not necessarily imply possible lung involvement, unlike CXR, hence our decision to use the latter.

3.A lot of data has been conveyed which look superfluous, Kindly rearrange with paragraphs and continuity with each other.

à We thank the Reviewer for this comment that has prompted us to rearrange some paragraphs as per your suggestion.

4.Same applies to statistical analysis 

We have also provided more details on our statistical analysis (please also see our reply to Reviewer 2)

5.Do you think a flowchart can be added so that lot of information can be transferred there and reduce tables.

à We thank the reviewer for his/her suggestion to present the CXR-score based clinical interpretation as a flowchart, unfortunately the Cox-PH regression analysis does not allow such interpretation. To establish a data driven flowchart, the appropriate analytic tool must be decision tree.

6.Few information about parameters of mechanical ventilation is missed like PEEP, mode ? used, FiO2 maintained during the process , patients who used MV for long are more prone for severe scores ?

à There were marked interindicidual fluctuations regarding the data derived from these parameters We had therefore to choosn most objective ending outcome, e.g.survival vsdeath, or ICU admission versus non admission  

7.Discussion 

Reduce the amount of words in Discussion Please discuss findings only 

Divide into paras so that they are easy to read 

Limitation to study ??

Were the results compared To MRI ,CT scores as it remains the Gold standard to check the severity of lung 

In the discussion, we have analyzed based on the results obtained from the study, explaining the selection of a comprehensive CXR score based on the extent and severity of lesion on radiograph, as well as the utility of CXR to screen for disease stratification for appropriate treatment instead of MRI, CT or ultrasound to limit the contamination on medical staff and especially in case of urgent epidemics in localities with limited medical access. We also shared the study's limitation in the last paragraph. As per the Reviewer’s suggestion, we have kept the words count to make our text concise and straight to the point. Hopefully these results can help the scientific community in general and public health planners, and especially clinicians in improving the effectiveness of healthcare.
